# Season, Terrestrial Ultraviolet Radiation, and Markers of Glucose Metabolism in Children Living in Perth, Western Australia

**DOI:** 10.3390/ijerph16193734

**Published:** 2019-10-03

**Authors:** Catherine L. Clarke, Lana M. Bell, Peter Gies, Stuart Henderson, Aris Siafarikas, Shelley Gorman

**Affiliations:** 1Telethon Kids Institute, University of Western Australia, Perth 6872, Australia; cathy.clarke@hotmail.com (C.L.C.); aris.siafarikasl@health.wa.gov.au (A.S.); 2Department of Paediatric Endocrinology and Diabetes, Perth Children’s Hospital, Nedlands 6009, Australia; lana.bell@health.wa.gov.au; 3Australian Radiation Protection and Nuclear Safety Agency (ARPANSA), Yallambie 3085, Victoria, Australia; peter.gies@arpansa.gov.au (P.G.); stuart.henderson@arpansa.gov.au (S.H.); 4Medical School, Division of Paediatrics, University of Western Australia, Crawley 6009, Australia; 5Institute for Health Research, University of Notre Dame, Fremantle 6160, Australia

**Keywords:** season, sunlight, terrestrial ultraviolet radiation, paediatric, blood glucose, insulin, HbA1c

## Abstract

Seasonality in glucose metabolism has been observed in adult populations; however, little is known of the associations between season and glucose metabolism in children. In this study, we examined whether markers of glucose metabolism (fasting glucose, insulin and HbA1c) varied by season in a paediatric population (6–13 years of age) located in Perth (Western Australia, *n* = 262) with data categorised by weight. Linear regression was used to analyse the nature of the relationships between mean daily levels of terrestrial ultraviolet radiation (UVR) (prior to the day of the blood test) and measures of glucose metabolism. Fasting blood glucose was significantly lower in autumn compared to spring, for children in combined, normal and obese weight categories. Fasting insulin was significantly lower in autumn and summer compared to winter for individuals of normal weight. HbA1c was significantly higher in summer (compared with winter and spring) in overweight children, which was in the opposite direction to other published findings in adults. In children with obesity, a strong inverse relationship (r = −0.67, *p* = 0.002) was observed for fasting glucose, and daily terrestrial UVR levels measured in the previous 6 months. Increased safe sun exposure in winter therefore represents a plausible means of reducing fasting blood sugar in children with obesity. However, further studies, using larger paediatric cohorts are required to confirm these relationships.

## 1. Introduction

Type-2 diabetes (T2D) is characterised by progressive insulin resistance, raised blood glucose levels and altered lipid metabolism that may result in macro- and micro-vascular diseases, such as cardiovascular disease, stroke, diabetic retinopathy, neuropathy and nephropathy [1,2]. Although usually considered a disease of adulthood, rates of T2D are rising in paediatric populations. For example, between 1990 and 2012, the annual incidence of T2D in children living in Western Australia rose more than 10-fold, from 0.2 to 3.1 per 100,000 person-years [3]. This burden is felt particularly by First Nation peoples, with the prevalence of T2D 8-times greater in children of Australian Indigenous, than non-Indigenous ethnic background (10–14 years of age) [4]. The rising incidence of paediatric T2D is associated with increasing rates of childhood obesity [5] with excessive caloric consumption and sedentary lifestyles [6] thought to be the main drivers. However, other environmental factors may contribute towards the development of T2D and obesity in children. We have recently hypothesised that one such environmental contributor in childhood may be reduced exposure to ultraviolet radiation (UVR) from sunlight [7].

UVR is part the solar spectrum, and can be divided into UVA (λ = 315–400 nm), UVB (λ = 280–315 nm) and UVC (λ = 100–280 nm) radiation, of which only UVA (95%) and UVB (5%) reach the Earth’s surface. We previously observed reduced fasting insulin and glucose levels in the blood of mice fed a high fat diet exposed to non-burning doses of UVR (compared to mock-irradiated control mice fed a high fat diet) [8]. There is also evidence that increased exposure to UVR may improve glucose metabolism and cardiometabolic outcomes in adults [7]. These observations suggest a role for ongoing sub-erythemal exposure to UVR in glycaemic control. In temperate climates, time of year or season can be used as a proxy for exposure to sunlight [9] (or UVR), whereby solar exposure is greatest in summer and lowest in winter. Seasonality in cardiometabolic outcomes has been observed in adult populations, often with improved outcomes reported in summer [7]. In diabetic adults, haemoglobin A1c (HbA1c) was lower in summer compared to winter [10,11]. Furthermore, fasting glucose, body mass index (BMI), and blood lipids in patients with coronary artery disease were greater in winter compared with summer, and correlated with higher incidence of acute myocardial infarctions in winter [12]. Seasonal variation in blood pressure has also been observed [13], with lower systolic blood pressure occurring in summer compared to the other seasons [14].

While most published studies suggest glycaemic control in adults is influenced by season (and potentially sun exposure), there is little known as to whether seasonal variation exists in children. Some inverse associations between higher circulating levels of 25-hydroxyvitamin D (a proxy for sun exposure) and lower fasting glucose, insulin and other markers of glucose metabolism have been observed in paediatric populations [15,16], suggestive of inverse associations between sun exposure and glucose metabolism. With adult clinical trials reporting only limited benefits of vitamin D supplementation for preventing T2D [17,18], it is important to re-consider sun exposure itself as an important environmental modifier of metabolic dysfunction. This is particularly important for children with obesity, who are at increased risk of T2D. We hypothesise that greater exposure to UVR in summer may improve glycaemic control in children, as observed in adults. Here, we investigated whether there were seasonal differences in blood-based markers of glucose metabolism (fasting glucose, insulin and HbA1c) in a community- and obesity clinic-sourced population of children (6–13 years of age) residing in Perth (Western Australia), comparing findings across seasons from single blood sample (per participant). We also examined the associations of these markers of glucose metabolism with recent exposure to sunlight, using terrestrial UVR measured prior to the blood test, as a proxy for sun exposure.

## 2. Methods

### 2.1. Study Population

Methods for the collection of data were approved by the Child and Adolescent Health Services Human Research Ethics Committee (approval number EP853). Investigations were carried out following the rules of the Declaration of Helsinki (1975). The study population consisted of children and adolescents (6–13 years of age) who participated in a previously published study with informed consent [19]. Recruitment occurred from randomly selected primary schools in Perth (Western Australia) (*n* = 163), or through an initial consultation for weight-related issues with a qualified paediatrician at Princess Margaret Hospital (Perth, Western Australia) (*n* = 99) [19] (Figure 1). Inclusion criteria for the current study were a result recorded for fasting glucose, insulin or HbA1c. Children with medical or genetic conditions which affected body weight (e.g., inflammatory bowel disease, Prader-Willi syndrome) were excluded [19]. Recruitment occurred between 2004 and 2009 [19]. Perth is a coastal city, located at 31.96 °S and 115.87 °E, and experiences a temperate climate. In summer and winter, mean daily solar exposure is 28.4 and 10.7 MJ/m^2^, and the mean daily sunshine hours are 10.4 and 5.5, respectively [20].

### 2.2. Physical Examination and Weight Classification

Upon entry to the study, anthropometric measurements were obtained. Weight was measured using a digital balance scale to 2 decimal places. Height was obtained using a wall-mounted stadiometer (Holtain) to the nearest millimetre. BMI was calculated as weight/(height)^2^. Children were classified as being of normal weight, overweight or obese by their BMI-adjusted for age and sex as per the standard definitions [21] with the numbers in each weight category shown in Figure 1.

### 2.3. Markers of Glucose Metabolism

Blood tests were performed at Princess Margaret Hospital for Children located in Perth (Western Australia). Each child supplied one blood sample upon recruitment. Following an overnight fast, blood was drawn for subsequent measurement of glucose, insulin and HbA1c levels. Blood glucose was determined using a spectrophotometric (colorimetric) method (Vitros Glu, Ortho-Clinical Diagnostics, NY, USA), plasma insulin by a chemiluminescent immunoassay (Immulite 2000 Diamond Diagnostics, MA, USA) [19] and HbA1c on a Roche Integra 800 (Roche Diagnostics, Mannheim, Germany). As per Figure 1, there was some data missing for these measures, with these participants excluded.

### 2.4. Season and Daily Terrestrial UVR

Seasons were defined as: summer = December to February; autumn = March to May; winter = June to August; and, spring = September to November. The Australian Radiation Protection and Nuclear Safety Agency collects daily total terrestrial UVR levels from a data logger located at Perth airport (31.9274 °S, 115.9758 °E). The data used for this study was collected from January 2003 to December 2009 and reported in standard erythemal doses (SED), where 1 SED is equivalent to 100 J/m^2^ of radiant exposure [22]. Missing data due to technical malfunctions of the data collector was substituted by the mean terrestrial UVR levels for the 3 previous and 3 following days. If the missing data were for greater than 7 days, data were substituted by the mean terrestrial UVR levels collected for that date in other years between 2003–2010. When values for known (or ‘non-missing’) levels of terrestrial UVR were predicted using this method, they did not vary by more than 21% of the known values. The mean daily terrestrial UVR levels were calculated for the previous week, 2 weeks, 4 weeks, 3 months and 6 months prior to each blood draw. Each participant provided a fasting blood sample on a single occasion (on one day of the year), with the local terrestrial UVR levels calculated for the previous 1-week to 6-months for that person. Therefore, the day of blood collection determined terrestrial UVR levels for each individual. These varying time periods were investigated due to the uncertainty and limited literature on the effect of short- or long-term UV exposure on glucose metabolism, particularly in children. The mean daily terrestrial UVR level for each season was calculated for all days within each season.

### 2.5. Statistical Analyses

All data were analysed using GraphPad Prism (v7.03, 2017, GraphPad Software, San Diego, CA, USA). We did not perform an initial power calculation for sample size, as we examined data already collected, and thus could not modify the sample size. Individuals were classified by weight (normal weight, overweight or obese) as per the criteria of [21]. A D’Agostino & Pearson normality test was used to determine whether data were normally distributed. For comparison of data between seasons, a one-way ANOVA with Tukey’s post-hoc (if normally-distributed), or Kruskal-Wallis test with Dunn’s post-hoc analysis (if not normally-distributed) was performed to determine whether there were statistical differences within (and between) each weight category for each glycaemic outcome (mean fasting blood glucose, insulin and Hba1c), unless otherwise stated. To further investigate the nature of the relationships between sun exposure and measures of glucose metabolism, we used daily mean terrestrial UVR levels in times preceeding the blood test, as a proxy for prior sun exposure. We calculated the mean daily terrestrial UVR levels (x-axis) for the previous week, 2 weeks, 4 weeks, 3 months and 6 months prior to date of the blood test for each individual. Linear regression was then used to assess the relationships between each measure of glucose metabolism and terrestrial UVR levels for each time period. The slope(ß) of each generated line-of-best fit represented the relationship between the metabolic measure (y-axis) and mean daily terrestrial UVR for each time interval (x-axis). A Spearman’s test for linearity was performed (as data were not normally distributed) with the correlation(r) and *p*-value calculated. For all tests, differences were considered significant at *p*-values < 0.05.

## 3. Results

### 3.1. Population

Data were collected from 262 children who had an average age of 9.7 years (age range = 6–13 years). The majority of the children were classified as normal weight (62.0%) compared to those classified as overweight (30.5%) or obese (7.3%) (Table 1). Similar proportions of females (53%) and males (47%) were recruited into this study. However, there were significantly more males in the obese category (Table 1, Fisher’s exact test: *p* = 0.009). There was no significant difference between the weight categories in blood fasting levels of glucose or HbA1c (Table 1). Fasting blood insulin levels were significantly greater for individuals in the obese category compared to individuals from the normal weight and overweight categories (Table 1, one-way ANOVA with Tukey post-hoc: *p* < 0.0001). Fasting insulin levels for those in the overweight category were greater than those observed in individuals of normal weight (Table 1, one-way ANOVA with Tukey post-hoc: *p* = 0.007). There was no significant difference between the weight categories in daily terrestrial UVR levels measured (for any time span) prior to the blood test (Table 1). Even though daily terresterial UVR levels were all obtained from a single data logger at the Perth airport, there was considerable variability across the seasons (see Table 1). The highest daily terrestrial UVR occurred in summer (59 ± 10.6 SED, mean ± SD) and the lowest in winter (13 ± 4.3 SED) (Table 1).

### 3.2. The Relationship between Season and Fasting Blood Glucose, Insulin and HbA1c.

Significant differences in blood levels of fasting glucose were observed, with levels lowest in autumn and greatest in spring for individuals from the combined, normal and obese weight categories (Figure 2A; *p* ≤ 0.03). There were also significantly lower levels of fasting blood glucose in autumn than in winter, in individuals of the combined weight category (Figure 2A; *p* = 0.008). No significant differences in fasting blood glucose were observed across the seasons in overweight individuals (Figure 2A). Significant seasonal differences in fasting insulin were only observed in individuals of the normal weight category, whereby fasting insulin was significantly greater in winter compared to summer (*p* = 0.039) and autumn (*p* = 0.008) (Figure 2B). For individuals from the combined and overweight weight categories, HbA1c levels were significantly higher in summer compared to winter (Figure 2C; *p* ≤ 0.002) and spring (*p* ≤ 0.01). For individuals of normal weight, there was a trend for HbA1c levels to be higher in winter compared to autumn (Figure 2C; *p* = 0.07). No significant seasonal differences in blood HbA1c levels were observed in individuals from the obese category.

We also examined whether there were seasonal differences in glycolytic markers according to the sex of participants. Overall, fasting blood glucose (but not insulin nor HbA1c) levels were higher in males compared to females (combined data-male, 4.79 ± 0.43, *n* = 122; female, 4.58 ± 0.45, *n* = 134; mean ± SD; unpaired *t*-test; *p* = 0.0001). This difference also observed in individuals who were overweight (male, 4.77 ± 0.44, *n* = 79; female, 4.58 ± 0.50, *n* = 82; mean ± SD; unpaired *t*-test; *p* = 0.014). When analysed by season, significant differences in fasting glucose and HbA1c were detected in females, with lower fasting glucose observed in autumn compared to spring (*p* = 0.015), and higher HbA1c in summer compared to winter (*p* = 0.031) (Figure 3). In males, similar (although non-significant) findings were observed for males for fasting glucose (autumn v spring, *p* = 0.095), and HbA1c levels were greater in summer compared to spring (*p* = 0.035).

### 3.3. Terrestrial UVR and Fasting Glucose, Insulin and HbA1c.

A strong negative correlation between fasting glucose and daily terrestrial UVR levels measured in the previous 6 months was observed in individuals of the obese weight category (*n* = 19) (Table 2A, *p* = 0.002, r = −0.67 (−0.87 to −0.30 95% CI)), such that every increase of 1 SED, fasting glucose decreased by 0.026 mmol/L (ß = −0.026). Significant but generally weak relationships were observed for individuals in the combined weight category (*n* = 262), between fasting blood glucose and mean daily terrestrial UVR levels of the previous 2 weeks, 4 weeks, 3 months and 6 months (Table 2A, *p* ≤ 0.036, −0.131 ≥ r ≥ −0.281), with the strongest significant correlation observed when daily terrestrial UVR levels were measured over the previous 6 months (Table 2A, *p* ≤ 0.0001, r = −0.281). For individuals in the normal weight category (n = 161), significant but generally weak relationships were observed between fasting glucose and mean daily terrestrial UVR levels of the previous 4 weeks, 3 months or 6 months (Table 2A, *p* < 0.036, −0.165 ≥ r ≥ −0.269), with the strongest correlation observed at 6 months (Table 2A, *p* = 0.0006, r = −0.269, ß = −0.008). There were no significant relationships observed between fasting glucose and mean daily terrestrial UVR levels for individuals in the overweight category.

For individuals from the combined (*n* = 256), and normal weight (*n* = 161) categories, weak negative correlations were observed between fasting insulin and daily terrestrial UVR levels measured in the previous week, 2 weeks, 4 weeks or 3 months (Table 2B: combined *p* ≤ 0.004, −0.177 ≥ r ≥ −0.208; normal weight *p* ≤ 0.014, −0.193 ≥ r ≥ −0.242). For those individuals in the overweight category (*n* = 77), significant weak negative correlations were observed between fasting insulin and daily terrestrial UVR levels measured in the previous week, 2 or 4 weeks (Table 2B: *p* ≤ 0.031, −0.246 ≥ r ≥ −0.257). In children who were obese (*n* = 18), there were no significant relationships observed between fasting insulin and daily terrestrial UVR levels (for any duration prior to the blood test).

Weak positive correlations were observed between blood HbA1c levels and daily terrestrial UVR levels measured in the previous 3 months for children in the combined, normal weight or overweight categories (Table 3: *p* ≤ 0.035, 0.17 ≤ r ≤ 0.288). Furthermore, for individuals in the combined weight category, significant positive relationships were observed between HbA1c and daily terrestrial UVR levels measured over the previous 2 or 4 weeks, or 6 months (Table 3: *p* ≤ 0.044, 0.129 ≤ r ≤ 0.149). No significant relationship was observed between blood HbA1c levels and daily terrestrial UVR levels measured in obese children.

## 4. Discussion

In this study, we examined the relationships between season and terrestrial UVR on blood markers of glucose metabolism in children living in Perth (Western Australia). We observed seasonal differences in fasting blood glucose levels with levels lowest in autumn and highest in spring. Similarly, we observed an inverse relationship between fasting blood glucose and mean daily terrestrial UVR levels, with a strong significant relationship observed in children with obesity when daily terrestrial UVR levels were measured in the 6 months prior to the blood test. In children of normal weight, fasting insulin was significantly higher in winter compared to summer and autumn. Similarly, inverse (but weak) associations were observed between fasting insulin and recent levels (last 1 to 4 weeks) of terrestrial UVR for children in the normal weight and overweight categories. In contrast, HbA1c levels were significantly higher in summer compared to winter in overweight children, and a weak positive correlation was observed between blood HbA1c and daily terrestrial UVR levels measured in the previous 3 months. Seasonal differences were similar when data were stratified by sex. Together, these findings suggest there are seasonal differences in glucose metabolism in children, which may be due to variations in exposure to UVR, although these associations do not demonstrate causality (as discussed below).

Our observations are consistent with other studies in adults, in which seasonal differences in fasting blood glucose have been observed. In a large study of healthy Chinese adults (*n* = 49,417) fasting blood glucose decreased by 0.3 mmol/L from winter to autumn [23]. A similar phenomenon was also observed in an American cohort of 4,541 healthy Caucasian adults, with fasting glucose 0.6 mmol/L lower in summer compared to winter [24]. Diabetic adults in Greece had 1.5 mmol/L lower levels of fasting blood glucose in August (summer) compared to February (winter) [10]. Combined with data from the current study, these findings suggest that there is seasonality in fasting blood glucose in both adult and paediatric populations with changes in terrestrial UVR a plausible mechanism for reducing fasting blood glucose. Preclinical studies suggest that this mechanism may involve the release of nitric oxide bioactivity following the exposure of skin to UVR [8].

One important difference between our paediatric data and other adult data, is that fasting blood glucose was lowest in summer and highest in winter in most of these adult studies, and not autumn and spring, as observed in children in the current study. This may be due to differences between paediatric and adult metabolism, physiology, seasonal behaviours (e.g., physical activity and/or sun exposure), and variations in the latitude/weather of the study locations. However, a strength of the study design of some adult studies was longitudinal measurement of outcomes such as fasting blood glucose across seasons [10,23], while only a single measurement of blood was acquired from the child participants in the current study. Other important considerations include the larger datasets analysed in some adult studies [10,23] compared to this smaller dataset.

As expected, fasting insulin levels were significantly higher in children who were overweight or obese compared to children of normal weight, potentially indicative of early signs of insulin resistance. However, there were only weak associations between fasting insulin, season and terrestrial UVR. Reports from the literature are also inconclusive as to whether fasting insulin changes with season [7]. For example, insulin levels measured during an oral glucose tolerance test were increased in autumn compared to spring (*n* = 29) in one study of non-diabetic adults [25]. However, in another study, no seasonal difference in fasting insulin was demonstrated in 100 adults with cardiovascular disease [26]. We observed that fasting insulin was greatest in winter (compared to summer and autumn) in children of normal weight, in the opposite direction to some adult studies [27]. It is important to note that fasting insulin may be a poor marker for insulin sensitivity as compared to the gold-standard euglycemic hyperinsulinemic clamp [28]. This test is not often used (particularly in children) because of its significant cost, time burden and invasiveness to patients.

HbA1c levels were significantly greater in summer (compared to winter and spring) in the total population and overweight children. Similarly, positive associations were observed between fasting HbA1c levels and daily terrestrial UVR levels. HbA1c is representative of the glycaemic control measured over the past 2–3 months [29] (defined for adults) and is commonly used to assess glycaemic control in patients with diabetes. In diabetic adults living in Greece (*n* = 638) significantly lower HbA1c levels were recorded at the end of summer compared to the end of winter (reduced by 0.5%) [10]. Similar findings were also observed in diabetic veterans living in the USA (*n* = 285,705) [11], and in young people (mean age of 13 years, *n* = 27,035) with type-1 diabetes living in Germany and Austria [30], with HbA1c levels 0.2% less in summer compared to winter in both studies. These results are not consistent with the findings of our study, and may have been due to age-specific and study design differences highlighted above.

Individuals who lack or have restricted endogenous glucose regulation, such as those with diabetes, may have increased susceptibility to exogenous regulation of glucose. Indeed, we hypothesize that some of the inconsistent findings across the weight groups may be explained by this theory. One observation that fits with this hypothesis was the strong association between terrestrial UVR and fasting blood glucose in children with obesity, a risk factor for T2D [5], which were weaker and/or non-significant for children in other weigt catagories. Furthermore, elevated fasting insulin levels observed in children with obesity were suggestive of impaired endogenous glucose regulation. Other inconsistent findings across the weight categories may be explained by the smaller subset of individuals with obesity, and significant variation observed for some measures, particularly fasting insulin and in summer, for individuals in this weight category, than that observed in other children.

A major limitation of the current study was the size of the paediatric cohort, which was relatively small, particularly for individuals classified as overweight (*n* = 80) or obese (*n* = 19). Therefore, for some outcomes this study was likely inadequately powered. Larger and better-powered studies are required to determine whether the relationships observed in our cohort are applicable to wider paediatric populations in Australia and other locations. Indeed, in another larger Perth-based paediatric study, seasonal differences in diagnosis of type-1 diabetes in 0–14 year-olds was observed [31]. Our study was also limited by the nature of the dataset, which was acquired from a community cohort of children recruited through random selection of primary schools, and through children attending an obesity clinic [19]. Therefore findings observed here may not be representative of the general paediatric population in Perth. Due to the research design and analyses undertaken, we cannot make causal inferences based on these findings. In addition, seasonality in glycolytic measures is likely to be multi-factorial. Our analyses were done using a previously published dataset for which some important potential confounders were not measured, including diet, physical activity and skin tone [19]. Our study was thus unable to account for confounding for these and other factors, such as cultural practices, and other environmental conditions (e.g., temperature, humidity). Additionally, season and terrestrial UVR are proxies for sun exposure. To further evaluate the effects of UVR on glucose metabolism, more accurate measures of sun exposure are required, which may include participants wearing personal dosimeters and completing sun diaries and questionnaires [32].

These new findings may help health professionals working with paediatric populations consider how to better manage glycaemic dysfunction, particularly in children with obesity and T2D. It may be important to consider time-of-year, as a potential modifier of blood glucose, and lifestyle and environmental factors that may change with season, including physical activity levels, sun exposure, and diet. Consideration is needed around how to best address the lifestyle requirements of children at-risk or living with obesity, especially during autumn and winter. Further research is needed to determine whether promoting outdoor physical activity and ‘safe’ levels of sun exposure could help reduce elevated blood sugar levels and improve glycemic control in children at-risk or living with T2D.

## 5. Conclusions

We observed seasonal differences, and inverse associations between terrestrial UVR levels and fasting blood glucose in children living in Perth (Western Australia). More research is required using larger and better-defined datasets to refine and better quantify the relationships between glucose metabolism, and, lifestyle and environmental factors that change according season (including sun exposure) in children taking into account the numerous confounding factors, such as temperature. An improved understanding of the relationship between season and glycaemic control may aid in the prevention of diabetes and its associated co-morbidities.

## Figures and Tables

**Figure 1 ijerph-16-03734-f001:**
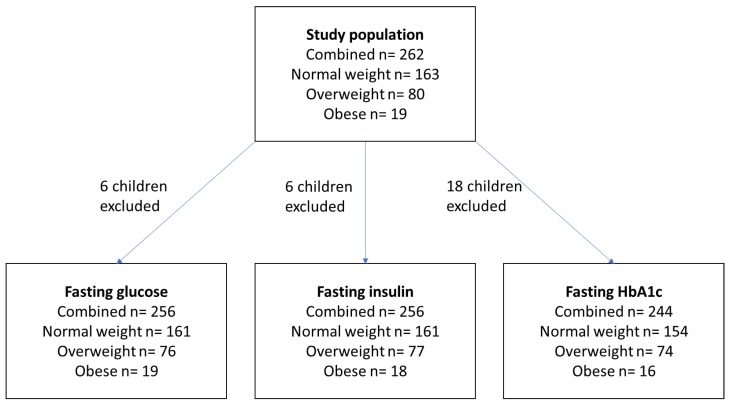
Participant numbers and exclusion for missing data. A total of 262 children were included in the study. In the analyses involving fasting glucose or insulin, 6 children were excluded, as they did not have a result recorded for fasting glucose or insulin. Another 18 children were excluded from analyses involving HbA1c, who did not have a result recorded for HbA1c.

**Figure 2 ijerph-16-03734-f002:**
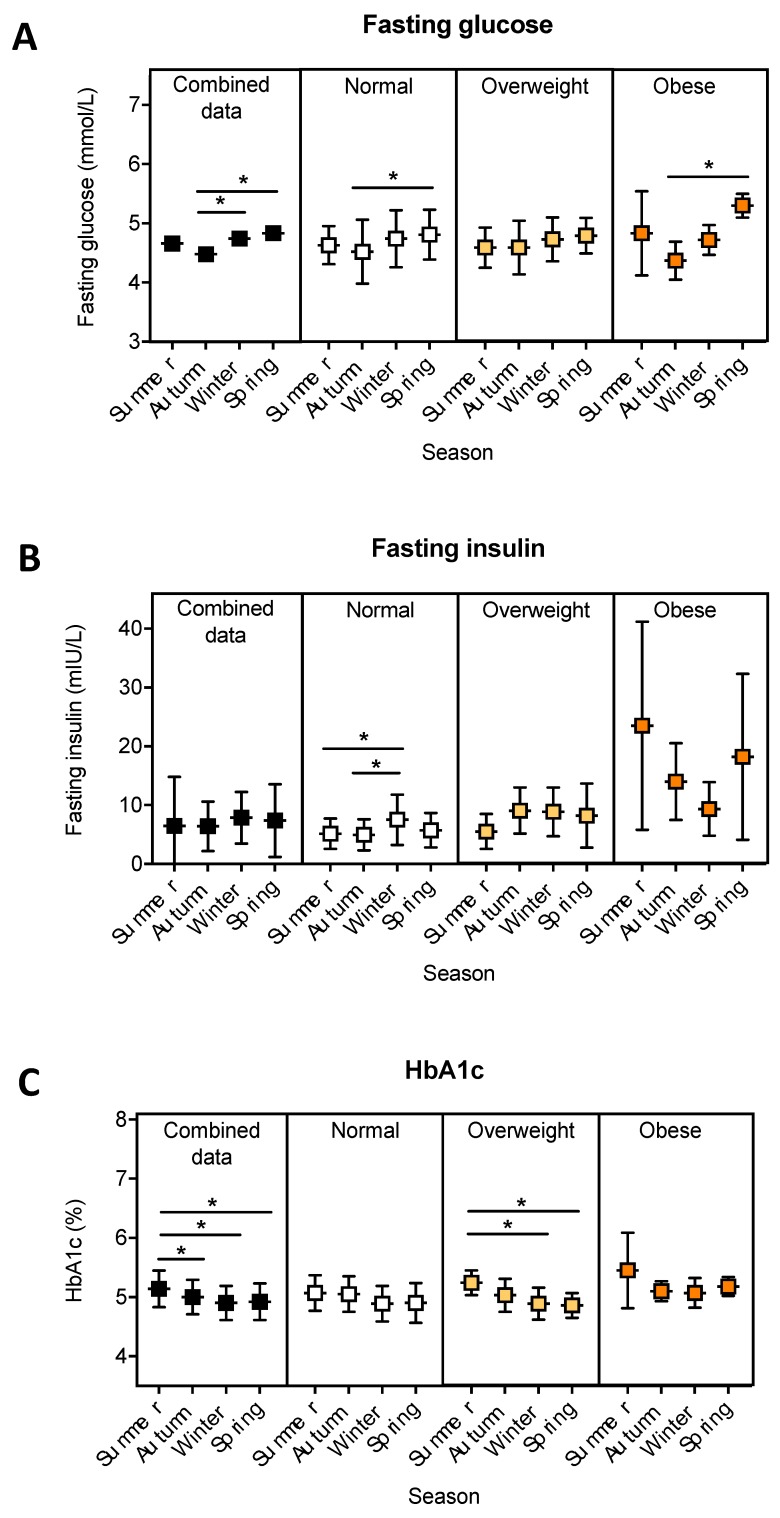
Fasting blood glucose, fasting insulin and HbA1c, according to season, and BMI. Fasting blood levels of glucose, insulin and HbA1c from a paediatric population (aged 6–13, *n* = 262) located in Perth (Western Australia) were categorised according to season of measurement, and weight status. Single measures of fasting blood glucose (*n* = 256), insulin (*n* = 256) and HbA1c (*n* = 244) were obtained. Individuals were categorised as being of normal, overweight or obese weight [21]. A one-way ANOVA with a Tukey’s post-hoc analysis was performed to test for differences between seasons. Data are shown as mean ± SD. Significant differences (*p* < 0.05) observed between seasons are denoted using an asterisk.

**Figure 3 ijerph-16-03734-f003:**
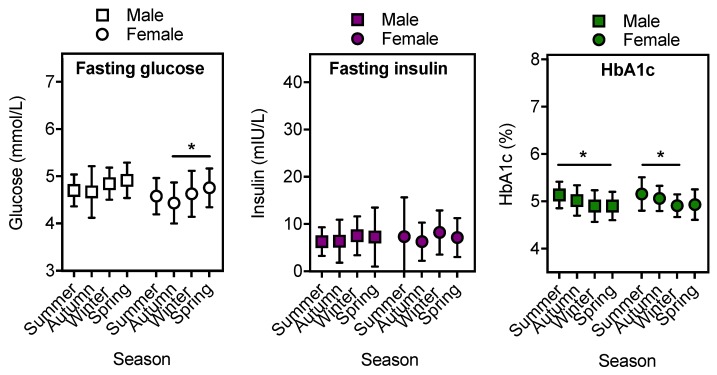
Fasting blood glucose, fasting insulin and HbA1c, according to season, and sex. Fasting blood levels of glucose, insulin and HbA1c from a paediatric population (aged 6–13, *n* = 262) located in Perth (Western Australia) were categorised according to season of measurement and sex (males = squares, females = circles). Single measures of fasting blood glucose (*n* = 256), insulin (*n* = 256) and HbA1c (*n* = 244) were obtained. One-way ANOVA with Tukey’s post-hoc analysis was performed to test for differences between seasons. Data are shown as mean ± SD. Significant differences (*p* < 0.05) are denoted using an asterisk.

**Table 1 ijerph-16-03734-t001:** General characteristics of the paediatric population. Shown below is the age, sex, glycaemic data and daily terrestrial levels of UVR (measured over the past week, 2 weeks, 4 weeks, 3 months and 6 months preceding the blood test) for the paediatric population residing in Perth (Western Australia) for the total population (combined data), and for data from individuals categorised as normal, overweight or obese (per an internationally-recognised assessment of childhood BMI [21]). The daily terrestrial UVR levels for each season are also shown.

Demographic Data	Combined Data	Normal Weight	Overweight	Obese
Total (*n*, %)	262	163, 62	80, 31	19, 7
Age^2^	9.7 ± 1.8	9.6 ± 1.9	9.8 ± 1.7	10.1 ± 1.5
Males (*n*, %)	122, 47	79, 48	30, 38	13, 68
Females (*n*, %)	140, 53	84, 52	50, 63	6, 32
Glycaemic data (mean ± SD)	Combined data	Normal weight	Overweight	Obese
Fasting glucose ^2^ (mmol/L)	4.7 ± 0.5	4.7 ± 0.5	4.7 ± 0.4	4.7 ± 0.5
Fasting insulin ^2^ (mU/L)	7.2 ± 5.0	6.0 ± 3.5	8.1 ± 4.3	14.7 ± 10.2
HbA1c ^2^ (%)	5.0 ± 0.3	5.0 ± 0.3	5.0 ± 0.3	5.2 ± 0.3
Daily terrestrial UVR^3^ (SED ^1^)	Combined data	Normal weight	Overweight	Obese
Past week ^2^	26.5 ± 15.6	27.3 ± 16.1	24.8 ± 14.5	27.6 ± 16.7
Past 2 weeks ^2^	26.6 ± 15.4	27.1 ± 15.4	25.7 ± 15.5	26.9 ± 15.9
Past 4 weeks ^2^	27.0 ± 15.6	27.3 ± 15.5	26.6 ± 16.0	26.7 ± 15.0
Past 3 months ^2^	30.2 ± 16.0	30.1 ± 15.5	31.0 ± 17.4	28.3 ± 13.6
Past 6 months ^2^	34.6 ± 12.6	34.2 ± 12.7	35.9 ± 12.7	32.9 ± 12.0
Daily terrestrial UVR ^3^ (SED)	Summer	Autumn	Winter	Spring
	59 ± 10.6	27 ± 12.6	13 ± 4.3	37 ± 12.8

^1^ SED = standard erythemal dose; ^2^ Data are shown as Mean ± SD; ^3^ For data collected between January 2003–December 2009, whereby local terrestrial UVR levels calculated for the previous 1-week to 6-months prior to the date of blood collection for each person. Further demographic data are reported elsewhere [5].

**Table 2 ijerph-16-03734-t002:** The associations between daily terrestrial UVR levels and fasting blood glucose or insulin. In a paediatric population (aged 6–13, *n* = 256) located in Perth (Western Australia), fasting blood glucose (**A**) and insulin (**B**) were measured. Individuals were categorised as being of normal, overweight or obese weight [21]. The results of the blood tests were plotted (y-axis) against the mean daily terrestrial UVR levels (x-axis, standard erythemal doses, SED) measured in the previous week, 2 weeks, 4 weeks, 3 months and 6 months before each blood test. A line of best fit was calculated and the slope (ß) of these linear relationships is shown ± standard error (SE). A Spearman’s test was used to test the significance of these relationships, with the correlation coefficient (r) and *p*-values shown. Significant relationships (*p* < 0.05) are shown in bold.

A. Fasting Blood Glucose (mmol/L).					
Terrestrial UVR (SED)		Combined *n* = 256	Normal Weight *n* = 161	Overweight *n* = 76	Obese *n* = 19
Previous week	**ß ± SE**	−0.003 ± 0.002	−0.003 ± 0.002	−0.006 ± 0.003	0.008 ± 0.007
	***p*-value (r)**	0.057 (−0.119)	0.094 (−0.132)	0.076 (−0.205)	0.296 (0.253)
Previous 2 weeks	**ß ± SE**	−0.004 ± 0.002	−0.004 ± 0.002	-0.006 ± 0.003	0.007 ± 0.007
	***p*-value (r)**	**0.036** (−0.131)	0.066 (−0.145)	0.113 (−0.183)	0.601 (0.128)
Previous 4 weeks	**ß ± SE**	−0.005 ± 0.002	−0.005 ± 0.002	−0.007 ± 0.003	0.004 ± 0.007
	***p*-value (r)**	**0.016** (−0.151)	**0.036** (−0.165)	0.095 (−0.193)	0.735 (0.0832)
Previous 3 months	**ß ± SE**	−0.007 ± 0.002	−0.007 ± 0.002	−0.007 ± 0.003	−0.010 ± 0.007
	***p*-value (r)**	**0.0003** (−0.226)	**0.003** (−0.234)	0.100 (−0.19)	0.117 (−0.372)
Previous 6 months	**ß ± SE**	−0.009 ± 0.002	−0.008 ± 0.003	−0.007 ± 0.004	−0.026 ± 0.007
	***p*-value (r)**	**<0.0001** (−0.281)	**0.0006** (−0.269)	0.069 (−0.210)	**0.002** (−0.672)
**B.** Fasting insulin (mU/L).					
**Terrestrial UVR** (SED)		**Combined** *n* = 256	**Normal Weight** *n* = 161	**Overweight** *n* = 77	**Obese** *n* = 18
Previous week	**ß ± SE**	−0.026 ± 0.020	−0.045 ± 0.017	−0.071 ± 0.033	0.240 ± 0.140
	***p*-value (r)**	**0.003** (−0.185)	**0.014** (−0.193)	**0.031** (−0.246)	0.117 (0.383)
Previous 2 weeks	**ß ± SE**	−0.027 ± 0.020	−0.048 ± 0.017	−0.064 ± 0.031	0.26 ± 0.15
	***p*-value (r)**	**0.003** (−0.186)	**0.009** (−0.206)	**0.024** (−0.257)	0.145 (0.358)
Previous 4 weeks	**ß ± SE**	−0.033 ± 0.020	−0.054 ± 0.017	−0.059 ± 0.030	0.24 ± 0.15
	***p*-value (r)**	**0.0007** (−0.208)	**0.002** (−0.242)	**0.030** (−0.247)	0.117 (0.383)
Previous 3 months	**ß ± SE**	−0.046 ± 0.020	−0.059 ± 0.017	−0.039 ± 0.028	0.076 ± 0.17
	***p*-value (r)**	**0.004** (−0.177)	**0.005** (−0.220)	0.319 (−0.115)	0.449 (0.190)
Previous 6 months	**ß ± SE**	−0.046 ± 0.025	−0.039 ± 0.022	−0.006 ± 0.039	−0.20 ± 0.20
	***p*-value (r)**	0.126 (−0.095)	0.084 (−0.137)	0.798 (0.030)	0.575 (−0.142)

**Table 3 ijerph-16-03734-t003:** The associations between daily terrestrial UVR levels and fasting blood HbA1c levels. In a paediatric population (aged 6–13, *n* = 244) located in Perth (Western Australia), fasting circulating HbA1c levels were measured. Individuals were categorised as being of normal, overweight or obese weight [21]. The results of the blood tests were plotted (y-axis) against the mean daily terrestrial UVR levels (x-axis, standard erythemal doses, SED) measured in the previous week, 2 weeks, 4 weeks, 3 months and 6 months before each blood test. A line of best fit was calculated and the slope (ß) of these linear relationships is shown ± standard error (SE). A Spearman’s test was used to test the significance of these relationships, with the correlation coefficient (r) and *p*-values shown. Significant relationships (*p* < 0.05) are shown in bold.

Terrestrial UVR (SED)		Combined *n*=244	Normal Weight *n* = 154	Overweight *n* = 74	Obese *n* = 16
Previous week	**ß ± SE**	0.002 ± 0.001	0.001 ± 0.002	0.004 ± 0.002	0.006 ± 0.004
	***p*-value (r)**	0.062 (0.120)	0.294 (0.085)	0.084 (0.202)	0.539 (0.165)
Previous 2 weeks	**ß ± SE**	0.002 ± 0.001	0.001 ± 0.002	0.004 ± 0.002	0.006 ± 0.004
	***p*-value (r)**	**0.027** (0.142)	0.204 (0.103)	0.086 (0.201)	0.450 (0.202)
Previous 4 weeks	**ß ± SE**	0.003 ± 0.001	0.002 ± 0.002	0.004 ± 0.002	0.006 ± 0.005
	***p*-value (r)**	**0.020** (0.149)	0.127 (0.124)	0.084 (0.202)	0.607 (0.138)
Previous 3 months	**ß ± SE**	0.003 ± 0.001	0.003 ± 0.002	0.004 ± 0.002	0.002 ± 0.006
	***p*-value (r)**	**0.003** (0.191)	**0.035** (0.170)	**0.013** (0.288)	0.968 (0.011)
Previous 6 months	**ß ± SE**	0.003 ± 0.002	0.003 ± 0.002	0.005 ± 0.003	−0.007 ± 0.006
	***p*-value (r)**	**0.044** (0.129)	0.097 (0.134)	0.062 (0.218)	0.339 (−0.254)

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
