# Peer review of "Season, Terrestrial Ultraviolet Radiation, and Markers of Glucose Metabolism in Children Living in Perth, Western Australia"

_ijerph, 2019, doi:10.3390/ijerph16193734_

Round 1

Reviewer 1 Report

Introduction:

Line 64-I think there needs to be an additional paragraph supported or at least a few more sentences on why is it important to study UVR exposure among children and make it clear what are the gaps in the existing literature and previous studies done on this topic. In the current version, it is not clear and that significance gets lost in the introduction.

Line 65-Provide references for the studies which have provided some evidence on this association.

Methods

Line 78-It would be good to provide the inclusion criteria of the participants here.

Line 80-Did the authors conduct a power calculation for the sample size? If not, why?

Results

It would help the readers to have a table on the demographic characteristics of the participants.

After line 162-It seems to me that the headings in the table columns are not aligned consistently. I would recommend the authors to check the formatting of the tables.

Discussion

It seems to me that findings of the study have important implications for research and practice. It would be good if the researchers can further elaborate on future research stemming from these findings.
How are these findings beneficial to health professionals working with pediatric populations at school or at home?

Limitation-This study, due to its research design and analyses, also looked at associations and therefore cannot make causal inferences based on the findings. 

The authors can make it clearer as to why they did not measure or control factors such as the diet and physical activity levels in their study (This could go in the limitations section).

Author Response

Point-by-point Response to Reviewers’ Comments for IJERPH-595688

Thank you for critiquing our submission. We have now modified the manuscript with new changes highlighted in blue text. Please find below a point-by-point response to the reviewers’ comments.

Reviewer 1:

Introduction - Line 64 - I think there needs to be an additional paragraph supported or at least a few more sentences on why is it important to study UVR exposure among children and make it clear what are the gaps in the existing literature and previous studies done on this topic. In the current version, it is not clear and that significance gets lost in the introduction.

A: We have added the following from line 64: “While most published studies suggest glycaemic control in adults is influenced by season (and potentially sun exposure), there is little known as to whether seasonal variation exists in children. Some negative associations between circulating levels of 25-hydroxyvitamin D (a proxy for sun exposure) and fasting glucose, insulin and other markers of glucose metabolism have been observed in paediatric populations[1,2], suggestive of inverse associations between sun exposure and glucose metabolism. With adult clinical trials reporting only limited benefits of vitamin D supplementation for preventing type-2 diabetes[3,4], it is important to re-consider sun exposure its self as an important environmental modifier of metabolic dysfunction. This is particularly important for children with obesity, who are at increased risk of type-2 diabetes.”

Introduction - Line 65 - Provide references for the studies which have provided some evidence on this association.

A: See our answer to Q1 above.

Methods - Line 78 - It would be good to provide the inclusion criteria of the participants here.

A: Inclusion criteria for the current study was a result recorded for fasting glucose, insulin or HbA1c. We have added this statement to line 87.

Methods - Line 80 - Did the authors conduct a power calculation for the sample size? If not, why?

A: We did not perform a power calculation for sample size, as we were examining data already collected, and thus were not able to modify the sample size for the timeframe for which data was collected (between 2004 and 2009). A statement has been added to the statistical analysis section to address this issue. (line 132)

Results - It would help the readers to have a table on the demographic characteristics of the participants.

A: In Table 1, we provide some general demographic data on age and gender for the ‘combined’ dataset, and also for the dataset stratified according to weight (normal, overweight, obese). Further demographic data is reported elsewhere[5]. We have added this point to the legend of Table 1.

Results - After line 162 - It seems to me that the headings in the table columns are not aligned consistently. I would recommend the authors to check the formatting of the tables.

A: The Table headings have been aligned.

Discussion - It seems to me that findings of the study have important implications for research and practice. It would be good if the researchers can further elaborate on future research stemming from these findings.

A: We have added a further sentence elaborating on future research that could arise from the findings of this paper (line 368). See also our response to point 8 below.

Discussion - How are these findings beneficial to health professionals working with paediatric populations at school or at home?

A: These new findings may help health professionals working with paediatric populations consider how to better manage glycaemic dysfunction, particularly in children with obesity and type-2 diabetes. It may be important to consider time-of-year, as a potential modifier of blood glucose, and quantify levels of recent sun exposure. As our findings suggest that some sun exposure may be necessary for reducing elevated fasting glucose levels in children with obesity, further consideration is needed around how to best balance the sun health requirements of children at-risk or living with obesity, especially during autumn and winter. This may be of particular importance for children with darker-skin (e.g. Type V-VI Fitzpatrick skin type), for whom sun exposure is less risky (for later-life skin cancers) than children with lighter-skin (e.g. Type I-II Fitzpatrick skin type). Further research is needed to determine whether promoting outdoor physical activity and ‘safe’ levels of sun exposure could help reduce elevated blood sugar levels and improve glycaemic control in children at-risk or living with type-2 diabetes. We have this paragraph to the Discussion section (from line 360).

Limitation - This study, due to its research design and analyses, also looked at associations and therefore cannot make causal inferences based on the findings. The authors can make it clearer as to why they did not measure or control factors such as the diet and physical activity levels in their study (This could go in the limitations section).

A: We could not measure or control for factors such as diet and physical activity, as these data were not collected in the original study. We have added statements to the Limitations section as suggested: “Due to the research design and analyses undertaken, we cannot make causal inferences based on these findings” (line 351) and, “Our analyses were done using a previously published dataset for which some important potential confounders were not measured, including diet, physical activity and skin tone[5]” (line 352).

Reviewer 2 Report

The manuscript is an original and very interesting study. However, the approach is difficult to understand since the authors mention that they performed the UVR measurements at the city's airport and that these results were the ones they used to determine the effect of this parameter on their population (if this is true) , I can't understand how values ​​change in different categories according to weight. In addition, there are other environmental variables that can affect the glycemic control of patients, for example; the physical activity, diet and consumption of any product that should be considered in the patient's questionnaire and that for some reason is not mentioned in the study. Therefore, authors are recommended to attach the relevant general information that may affect their study population and the interpretation of the results. Finally, in my opinion, figure 2 and table1 show the same information and it is not necessary to repeat the information in the writing, so it is suggested that the authors keep the table and append the statistical significance in the data.

Author Response

Point-by-point Response to Reviewers’ Comments for IJERPH-595688

Thank you for critiquing our submission. We have now modified the manuscript with new changes highlighted in blue text. Please find below a point-by-point response to the reviewers’ comments.

Reviewer 2:

The approach is difficult to understand since the authors mention that they performed the UVR measurements at the city's airport and that these results were the ones they used to determine the effect of this parameter on their population (if this is true), I can't understand how values ​​change in different categories according to weight.

A: We apologise for a lack of clarity in our approach. While the UVR measurements were all obtained from a single data logger at the Perth airport, there was considerable variability in levels of terrestrial UVR across the seasons (see Table 1), and throughout the year. Each participant provided a fasting blood sample on a single occasion (on one day of the year), with the local terrestrial UVR levels calculated for the previous 1-week to 6-months for that person. Therefore, the day of blood collection determines terrestrial UVR levels for each individual, who was then categorised according to body weight in Table 1. We have added clarifying statements to the methods (line 125) and results (line 160) and the legend of Table 1.

In addition, there are other environmental variables that can affect the glycemic control of patients, for example; the physical activity, diet and consumption of any product that should be considered in the patient's questionnaire and that for some reason is not mentioned in the study. Therefore, authors are recommended to attach the relevant general information that may affect their study population and the interpretation of the results.

A: As stated for our answer to Q9 of Reviewer 1, we could not measure or control for factors such as diet and physical activity, as these data were not collected in the original study that generated the dataset analysed. We have added a statement to the Limitations section: “Our analyses were done using a previously published dataset for which some important potential confounders were not measured, including diet, physical activity and skin tone[5].” (line 353)

Finally, in my opinion, Figure 2 and Table 1 show the same information and it is not necessary to repeat the information in the writing, so it is suggested that the authors keep the table and append the statistical significance in the data

A: While there are similarities in data presented in this table and figure, there are notable and important differences. In Figure 2 the glycaemic data from Table 1 has been separated by season. As seasonal comparison is an important aim of this study, and because of the complexity of the data we would prefer to keep Figure 2. Furthermore, a figure perhaps more clearly demonstrates the differences between seasons, particularly in the children with obesity, which helps the reader comprehend and more directly visualise the effect size and variability between and within weight categories as depicted in Figure 2.
